# Fate of Entosis: From the Beginning to the End in Untreated Advanced Breast Cancer

**DOI:** 10.3390/ijms241512142

**Published:** 2023-07-29

**Authors:** Ireneusz Dziuba, Agata M. Gawel, Paweł Tyrna, Jolanta Rybczynska, Lukasz P. Bialy, Izabela Mlynarczuk-Bialy

**Affiliations:** 1Department of Pathology, Faculty of Medicine, Academy of Silesia, 40-555 Katowice, Poland; 2Histology and Embryology Students’ Science Association, Department of Histology and Embryology, Faculty of Medicine, Medical University of Warsaw, 02-004 Warsaw, Poland; agata.gawel@wum.edu.pl (A.M.G.); pawel.tyrna@gmail.com (P.T.); 3Department of Histology and Embryology, Faculty of Medicine, Collegium Medicum, Cardinal Stefan Wyszynski University in Warsaw, 01-815 Warsaw, Poland; j.rybczynska@uksw.edu.pl; 4Department of Histology and Embryology, Faculty of Medicine, Medical University of Warsaw, 02-004 Warsaw, Poland; lbialy@wum.edu.pl

**Keywords:** homotypic entosis, cell-in-cell, breast cancer, cancer metastasis, tumor cell invasion, tumor cell survival, E-cadherin, Ki67

## Abstract

Homotypic entosis is a phenomenon in which one cancer cell invades a neighboring cancer cell and is closed entirely within its entotic vacuole. The fate of entosis can lead to inner cell death or survival. Recent evidence draws attention to entosis as a novel prognostic marker in breast cancer. Nevertheless, little is known about the quantity and quality of the process of entosis in human cancer specimens. Here, for the first time, we analyze the frequency of entotic figures in a case of NOS (Non-Other Specified) breast cancer with regard to location: the primary tumor, regional lymph node, and distant metastasis. For the identification of entotic figures, cells were stained using hematoxylin/eosin and assessed using criteria proposed by Mackay. The majority of entotic figures (65%) were found in the lymph node, 27% were found in the primary tumor, and 8% were found in the far metastasis. In the far metastases, entotic figures demonstrated an altered, atypic morphology. Interestingly, in all locations, entosis did not show any signs of cell death. Moreover, the slides were stained for E-cadherin or Ki67, and we identified proliferating (Ki67-positive) inner and outer entotic cells. Therefore, we propose additional criteria for the identification of pro-survival entotic structures in diagnostic histopathology.

## 1. Introduction

Entosis is a process of regulated cell invasion, in which one cancer cell invades into a neighboring cancer cell, resulting in the formation of cell-in-cell (CIC) structures [1]. Homotypic entosis involves two cells of the same type and is regulated by molecular mediators. Depending on the cell type, homotypic entosis typically lasts 6–12 h [2]. The fate of the inner cell in entosis can either be cell survival (“vital” entosis) or death (“lethal” entosis) [1,2]. Entosis has been reported in numerous cancers [2,3,4,5,6,7,8,9,10,11,12,13]. Breast cancer (BC) seems to be especially entosis-competent and entotic figures have been detected in both primary tumors, as well as metastatic lesions [6,7,8,9,10,14,15]. Entosis correlates with a more malignant phenotype, which is characterized by poorer prognosis and survival in some cancers [8,13,14,15]. Nevertheless, other existing data demonstrate that entosis may be associated with better prognosis in comparison to similar cases without entosis [16]. In BC, entosis was shown to be an independent prognostic factor, with the prognosis depending on the cancer subtype [15], correlating with HER2 and Ki67 expression [17], and with the highest prognostic value in young women [8].

Various molecular markers that play a role in entosis have been discovered. In the first step, after release from the substrate, one cell adheres to another cell via E-cadherin. Subsequently, the Rho/ROCK pathway is activated and triggers the formation of actomyosin filaments in the invading (the inner and stiffer) cell to push it into the outer and softer one [1,2].

Entosis can be initiated by starvation, certain chemotherapeutics, or aberrant mitosis [3,4,5,6]. It was shown that cells less likely to survive (with depleted AMP-activated protein kinase) are engulfed by the more metabolically competent ones [3]. Moreover, it has been postulated that stressful conditions may provoke lethal entosis by lysosomal digestion of the inner cell, as demonstrated on MCF7 cells in non-adherent conditions [1]. However, fully adherent conditions may rather lead to the vital form of entosis, as we demonstrated in the BxPC3 pancreatic cancer cell line [2].

Overall, studying entosis in routine histopathological examinations is difficult and time-consuming, as there are no automatic methods nor specific markers that can be used to assess entotic figures. There are no universal histopathological criteria for the identification of entosis. Mackay’s criteria [12] for entotic structures are the only ones that are recognized and applied by many laboratories. Entotic figures are countable in hematoxylin/eosin-stained (H&E) specimens, where at least four of the following six features are unambiguously identifiable: (i) the nucleus of the internalized cell; (ii) cytoplasm of the internalized cell; (iii) a crescent-shaped nucleus of the engulfing cell; (iv) cytoplasm of the engulfing cell; (v) the nucleus of the host cell; (vi) an entotic vacuole between the engulfed and the outer cell. Based on staining and the above criteria, entotic figures can be distinguished from cancer cells located in capillaries or macrophages, and from emperipolesis or other CIC processes [2,17,18]. However, studying the quality of entotic cells in histopathological specimens, in order to address the question whether the inner cell of the entotic structure is vital or lethal, is still challenging. Therefore, new reports concerning entosis are of great importance and bring novel data into this unexplored area.

Thus, we have analyzed entosis in specimens obtained from a BC patient untreated with any chemical medications. The study was performed on archival paraffin tissue specimens from a 75-year-old Caucasian woman diagnosed with double-negative, HER2-positive breast cancer pT4N3 at the time of the primary diagnosis of BC, primarily located in the right mammary gland. At the time of the first diagnosis, a primary metastasis to the lymph node in the right armpit was detected. The patient only agreed to surgical treatment. The radical surgery was performed to remove the right breast with the tumor and the local nodule metastasis. All tissues were resected with margins of healthy tissue. After the operation, the patient refused any form of chemotherapy, biological therapy, or radiation therapy. The recurrence of the disease, which involved the left breast and the left axillary fossa was diagnosed 16 months after the first surgery (Figure 1). The patient once again refused any form of treatment other than surgery. A radical surgery and excision of cancerous tissue with margins were performed. The patient died 8 months after the second surgery of cancer recurrence.

Within this study, recurrent tumors are compared to the primary tumor and lymph node metastasis of this patient.

## 2. Results

### 2.1. Descriptive Analysis of Entotic Figures in Breast Cancer

Entotic hotspots in the H&E-stained specimens were identified according to Mackay’s criteria, and subsequently marked and analyzed. Entotic structures were identified in hotspots in the middle part of the primary tumor (T), and in the middle region of both the primary lymph node metastasis (N) and the late contralateral axillary lymph node metastasis (M). All entotic figures demonstrated at least four out of the six criteria defined by Mackay. The nuclei of the inner cells were rounded, with visible heterochromatin and nucleoli, and all had a preserved nuclear envelope. In most structures, the entotic vacuole was visible, clear, and free of cytoplasmic material. 

In the primary lesion (T), regions enriched in entotic structures were localized neither at the border of the tumor nor in the pale, fibrotic regions (Figure 2). Most entotic figures were found in lowly-differentiated areas, rich in cells, where cellular composition did not form tubules or other organotypic structures.

At the time of diagnosis, metastasis to a local lymph node was detected (corresponding to “N” in the TNM classification system). Entotic structures were identified and analyzed in the lymph node metastasis (Figure 3). Hotspots positive for entotic figures were revealed to be localized in the middle parts of the lymph node and CIC structures were not found in organotypic structures. The entotic figures demonstrated similar morphology and cellular composition as in the primary tumor (see: Figure 2).

Images from the BC recurrence to the contralateral axillary lymph node are shown in Figure 4 (indicating “M” in the TNM classification system). In the recurrent metastasis, cancer cells demonstrated a different morphology. Cells were more atypical, i.e., cell nuclei contained more heterochromatin, and the area of cytoplasm was reduced. Despite the different cell appearance, the histological structure of the identified entotic cells did not differ from analogical structures in the previously described samples.

### 2.2. E-Cadherin Expression in the Tested Specimens

In the hotspots defined above, we also analyzed entotic structures by staining for E-cadherin (an epithelial marker) to exclude other types of CIC structures, such as emperipolesis. All entotic structures were E-cadherin positive (Figure 5). This type of staining made the identification of entotic figures more difficult, but Mackay’s criteria (at least four out of the six characteristics for recognition of entosis) could still be applied.

### 2.3. Ki67-Positive Entotic Figures

As described above, entosis in this patient did not induce degradation of the involved cells. Therefore, we decided to investigate whether they were positive for Ki67, a marker of proliferating cells. As shown in Figure 6, we identified both outer and inner Ki67-positive entotic cells.

### 2.4. Quantification of Entotic Figures in TNM Localizations

Since the presence of entotic figures can predict a worse prognosis in cancer, we aimed to investigate the density of entotic structures in all the examined specimens. In fields with entotic hotspots, the percentage of cells in entosis among all cancer cells was counted. Non-BC cells in histological images (such as fibroblasts, lymphocytes, or blood vessels) were omitted during counting. In the studied fields, at least 10,000 cells in total were analyzed and among these cells, the percentage of entotic figures was determined. The percentages of entotic figures were 2.4% (T), 5.8% (N), and 0.75% (M). As displayed in Figure 7, there was a significant increase (2.4×) in the yield of entotic figures in the lymph node metastasis (N) as compared to the primary tumor. However, in the far metastasis (M), the number of CICs was significantly reduced (3.2×) in comparison to the primary tumor (T). If we equate the entire number of entotic figures in TNM stages to 100%, the majority of entotic figures (65%) were found in the lymph node (N), 27% in the primary tumor (T), and 8% in the far metastasis (M).

## 3. Discussion

One of the poorly addressed questions concerns the frequency of entosis in different stages of cancer [7,8,13,19,20]. Previous studies comparing the density of entosis in primary cancer vs. metastasis revealed that CIC structures were favorable prognostic factors for local recurrence-free survival and disease-free survival in BC [8]. Nevertheless, entosis-positive patients had an unfavorable prognosis in regard to metastasis-free survival, with the highest prognostic value in young patients [8]. Schwegler et al. were the first to study the incidence of entosis in correlation to survival and prognosis on the example of nasopharyngeal cancer [13]. The group drew intriguing conclusions: cell-in-cell events were found to predict patient outcome in various types of head-and-neck cancer better than apoptosis and proliferation. This suggested that identification of entotic figures in cancer specimens might be used to guide treatment strategies [13].

Here, we report for the first time the frequency of entotic structures in a primary BC lesion (T from TNM), the lymph node metastasis (N from TNM) and the far metastasis (M from TNM) from the disease relapse. Until now, there have been no data on entosis in BC far metastases. Moreover, the studies on entosis lack well-recognized international diagnostic criteria that could be universally used to identify and describe this process. Initially, we assumed that the highest density of entotic figures would be observed in the far metastasis, corresponding to a more advanced cancer. However, our findings did not support this hypothesis, as the number of entotic figures in the far metastasis (M from TNM) was the lowest out of all studied locations. Additionally, we observed a significant decrease in the frequency of entosis in the terminal phase of BC (Figure 7), which stays in contrast to previous studies that reported that the degree of malignancy in general correlates with the number of entotic figures.

The fate of entosis in cell culture is either cell death or cell survival within the entotic vacuole [1]. Cell death is accompanied by the activation of digestive enzymes in lysosomes that leads to inner cell degradation and characteristic regressive changes [1]. The entotic figures observed in all three TNM localizations (Figure 2, Figure 3 and Figure 4) demonstrated no signs of cell degeneration or regressive changes, neither in the outer, nor in the inner cell. This observation can support the hypothesis of viable (pro-survival) entosis in BC. However, cells in entotic figures from the far metastases (M) (Figure 4) demonstrated a different morphology from that observed in the T or N locations (Figure 2 and Figure 3). They were more atypical, which suggests the different nature of this process during terminal stages of BC. It can be considered that such cells may harbor novel biological features and, e.g., require limited nutrients.

Recently, entotic figures were shown to correlate with expression of the human epidermal growth factor receptor 2 (HER2) and Ki67 (a common cell proliferation marker) in BC [17,21,22]. As the number of Ki67-positive cells (30%) was constant in all tumor locations from the TNM classification, the correlation between entosis and Ki67 is not clear. In the present study, we were able to find both internal and external Ki67-positive cells in all TNM locations (Figure 6). This observation strongly supports the hypothesis that the fate of entosis in BC is not cell death, but rather cell survival. We propose that usage of Ki67 may be considered as an additional marker for viable entosis, especially since this staining is commonly used in histopathological diagnosis of cancer. In contrast we do not recommend counting entotic figures on slides stained for E-cadherin, which is generally used as a marker of the epithelial phenotype and cell-to-cell adhesion. Although it can increase the precision of entotic figure counting by distinguishing entotic figures from other CIC structures (such as emperipolesis), the strong membranous staining makes the observation of entotic figures more difficult (Figure 5).

Overall, we have demonstrated that Mackay’s criteria are valuable for entotic screening, and analysis can be based on classical H&E staining. It makes this test affordable and most importantly, unambiguous, in the detection of entotic cells. We additionally propose the following histopathological criteria for the identification of pro-survival entosis: **(1) a clear entotic vacuole; (2) a preserved inner cell nucleus; (3) visible nucleoli; (4) visible heterochromatin and (5) a clear margin of the cytoplasm of the inner cell.** At least three out of the five above criteria should be met to classify a structure as a viable entosis. Based on our studies, we propose Ki67 as an additional staining for the recognition of pro-survival entosis, in order to demonstrate Ki67-positive inner entotic cells.

## 4. Materials and Methods

**Histopathological procedures:** The material underwent standard diagnostic histopathologic procedures with H&E staining and immunohistochemistry analysis using the DAKO EnVision™+ System (Dako; Carpinteria, CA, USA). For the purpose of this study, the slides were immunostained for E-cadherin (an epithelial marker; Roche Diagnostics; Rotkreuz, Switzerland) and Ki67 (a marker of cell proliferation; Roche Diagnostics; Switzerland).

The immunohistochemistry procedures were performed as described in [17]. The controls were conducted for each slide as follows: the negative controls were: E-cadherin - lobular breast cancer and for Ki67 - skin. The positive controls were: E-cadherin - NOS breast cancer; Ki76 - small-cell lung cancer and palatine tonsil. All controls were performed with the same antibodies used within this study. The types and clone numbers of antibodies are given in [17].

Histopathological diagnosis: invasive breast cancer NOS, NST NHG 3 (3+3+3)/28 mitoses/10HPF ER−, PR−, HER2+++; Ki67-positive in 30% of nuclei. Both the primary breast lesion (T) and local lymph nodes (N) as well as secondary metastasis (M) were obtained from the same case. The study was approved by the bioethical committee. The full clinical and histopathological data on this patient, including macroscopic image and follow-up of the treatment, were also available. After the macroscopic and microscopic examination, and the histopathological diagnosis, an additional analysis of entotic figures and their frequency was performed. The slides were scanned using an Aperio GT 450 histological scanner (Leica; Wetzlar, Germany).

The histological scans containing >10,000 cells were assessed by three independent medical doctors, including a certified clinical pathologist. The number of entotic figures was counted using diagnostic criteria established by Mackay [12]. Only figures displaying at least 3 of the 6 diagnostic criteria and confirmed by three medical doctors were calculated.

We analyzed at least 40 randomly captured images in each of the TMN localizations. Average number of cells per image was 400 (T) and 250 (NM). Thus, we analyzed at least 10,000 cells in each of the TMN localizations.

The data analysis was performed using the GraphPad Prism 6.0 (GraphPad, San Diego, CA, USA) software. For statistical purposes, the one-way Anova was used. Statistical significance was considered at *p*-value < 0.01.

**Histopathological report**:

**Histopathological description** of T and N. Invasive cancer NOS, NST NHG 3 (3+3+3)/28 mitoses/10 HPF (field diameter 0.55 cm) and lymphangitic carcinomatosa. Ductal carcinoma in situ foci with high nuclear atypia, constituting about 10% of the cross-sectional area of the tumor, were also present. Extensive foci of necrosis and fibrosis. The tumor infiltrates the skin of the mammary gland, causing ulceration. Pathological classification of the tumor according to pTNM: pT4N3M0.

**Histopathological description of the relapse (M):** Metastasis of carcinomatous lymph nodes No III/XIX. Cancerous infiltration of the capsule of the lymph nodes and perinodal adipose tissue. In the upper outer quadrant, foci of atypical intraductal hyperplasia were also present. Pathological classification of the tumor according to pTNM: pT4N3M1. For macroscopic image see Appendix A.

## Figures and Tables

**Figure 1 ijms-24-12142-f001:**
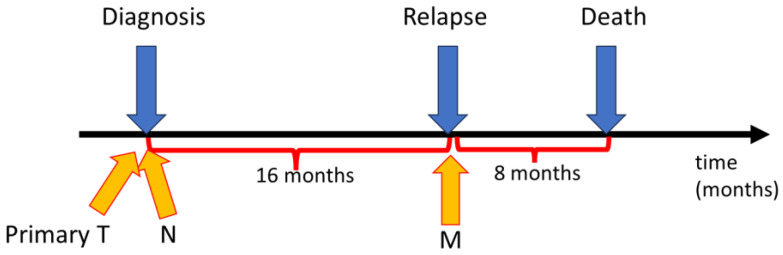
The course of stages of the described breast cancer (BC) case on a timeline. Yellow arrows indicate when tissues were taken for further histopathological examination. T—primary tumor, N—nodule infiltration by cancer cells, M—far metastasis.

**Figure 2 ijms-24-12142-f002:**
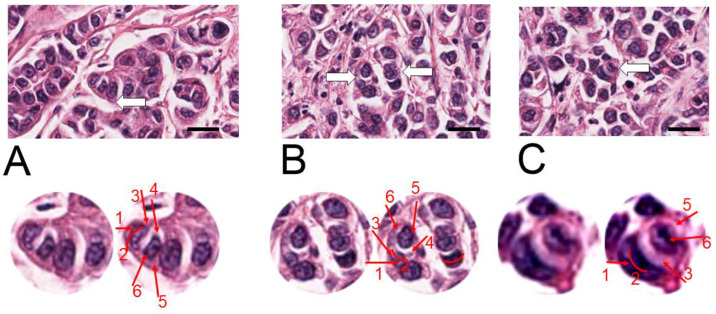
H&E staining of the primary cancer lesion (T). The three representative hotspots (**A**–**C**), with entotic figures are visualized (**upper panels**). The white arrows indicate selected entotic figures, which are shown in the **lower panels**. The entotic structures are digitally enlarged and displayed in duplicates: raw image (images on the left) and with indication of all visible Mackay’s criteria (on the right). Numbers 1–6 as follows: (1,2) semi-crescent nucleus of the engulfing cell; (3) cytoplasm of the outer cell; (4) entotic vacuole; (5) cytoplasm of the inner cell; (6) nucleus of the internalized cell. Scale bar—upper images 20 µM. (For whole slide image see Appendix A).

**Figure 3 ijms-24-12142-f003:**
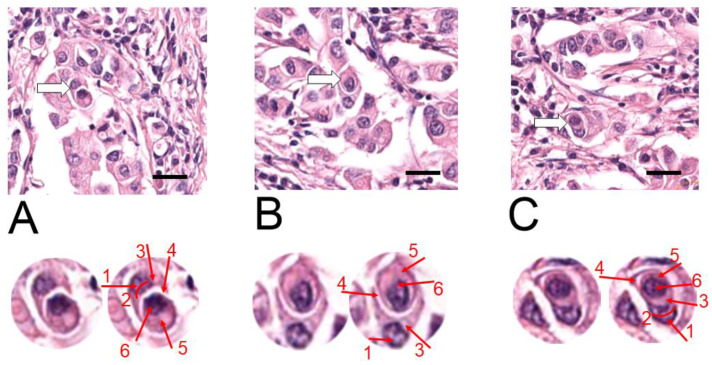
H&E staining of the lymph node metastasis (N). The three representative hotspots (**A**–**C**), with entotic figures are visualized (**upper panels**). The white arrows indicate selected entotic structures, which are shown on the **lower panels**. The entotic structures are digitally enlarged and displayed in duplicates: raw image (images on the left) and with indication of all visible Mackay’s criteria (on the right). Numbers 1–6 as follows: (1,2) semi-crescent nucleus of the engulfing cell; (3) cytoplasm of the outer cell; (4) entotic vacuole; (5) cytoplasm of the inner cell; (6) nucleus of the internalized cell. Scale bar—upper images 20 µM. (For whole slide image see Appendix A).

**Figure 4 ijms-24-12142-f004:**
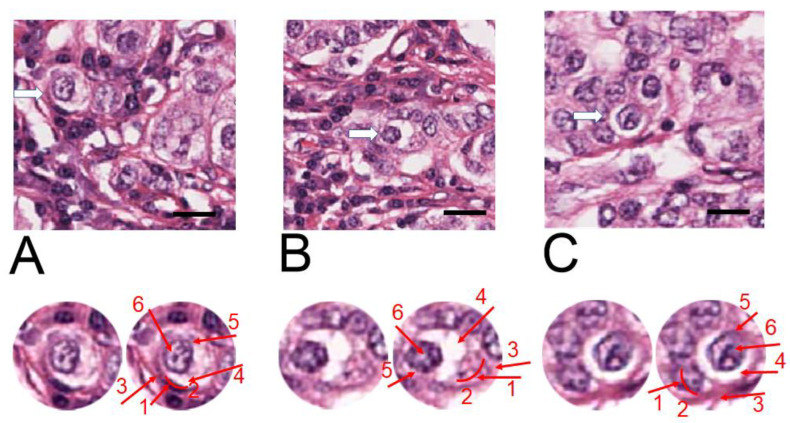
H&E staining of the contralateral axillary lymph node far metastasis (M). The three representative hotspots (**A**–**C**), with entotic figures are visualized (**upper panels**). The white arrows indicate selected entotic structures, which are enlarged and displayed in duplicates below (**lower panels**). Raw image (two images on the left) and with indication of all visible Mackay’s criteria (on the right). Numbers 1–6 as follows: (1,2) semi-crescent nucleus of the engulfing cell; (3) cytoplasm of the outer cell; (4) entotic vacuole; (5) cytoplasm of the inner cell; (6) nucleus of the internalized cell. Scale bar—upper images 20 µM. (For whole slide image see Appendix A).

**Figure 5 ijms-24-12142-f005:**
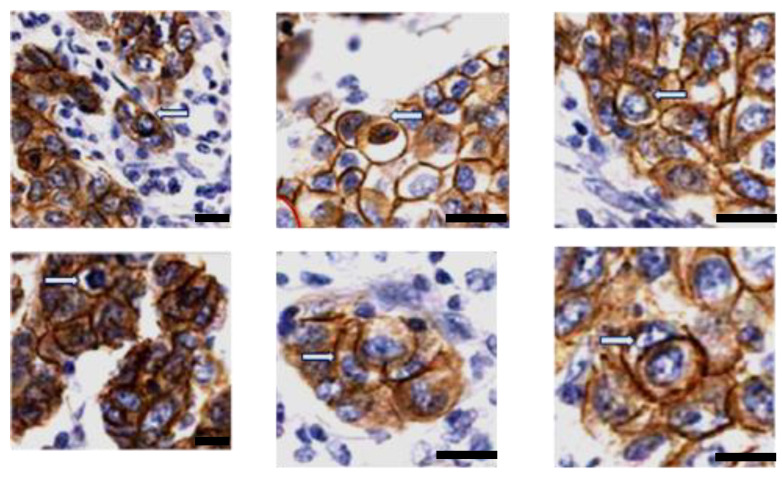
E-cadherin expression. Entotic cells are E-cadherin positive. On each panel, a representative entotic structure is indicated (white arrow), showing that both the inner and the outer entotic cell is of epithelial origin. Scale bar 20 µM.

**Figure 6 ijms-24-12142-f006:**
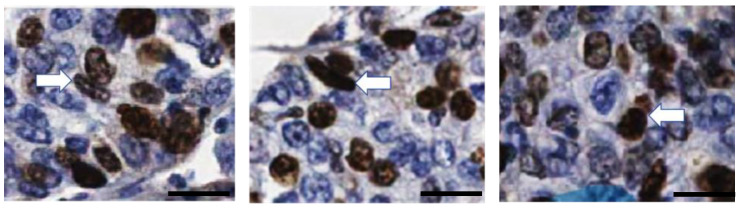
Ki67-positive entotic cells. Representative entotic structures are indicated using the white arrows. Both the inner and outer cell in the entotic structure is Ki67-positive. Scale bar 20 µM.

**Figure 7 ijms-24-12142-f007:**
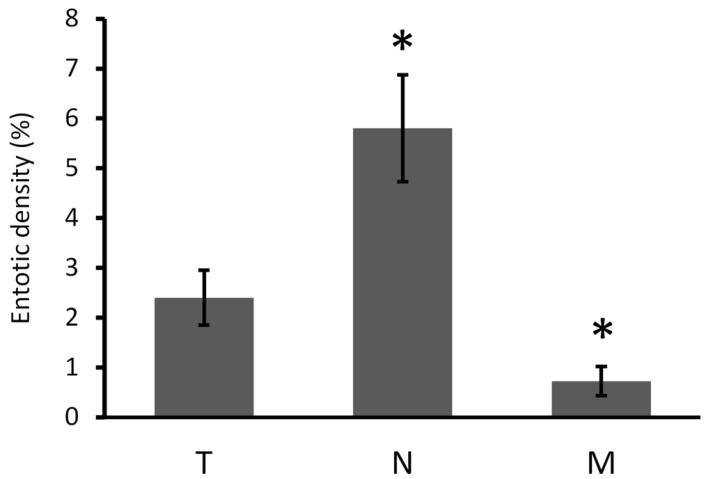
The density of entosis (frequency in %) in different localizations of the BC case according to the TNM criteria. T-tumor, N-lymph node, M-metastasis. The calculated density for each TMN localization was 2.4% (T), 5.8% (N), and 0.75% (M) of entotic figures. Bars represent SEM. The one-way Anova was used in the analysis; * *p* < 0.001.

## Data Availability

Data are available on request due to restrictions, e.g., privacy or ethical considerations. The data presented in this study are available on request from the corresponding authors.

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
