# Peer review of "Fate of Entosis: From the Beginning to the End in Untreated Advanced Breast Cancer"

_ijms, 2023, doi:10.3390/ijms241512142_

Round 1

Reviewer 1 Report

In their manuscript „Fate of entosis: from the beginning to the end in untreated advanced breast cancer“, I. Dziuba and colleagues investigated formaline fixed parrafine embedded tisuue samples of a woman diagnosed with advanced breast cancer. The study revealed new knowledge in the field of entosis. In it’s current form it is not suitable for publication though. Beside some minor issues concerning wording, grammar and abbreviations, there are some major concerns:

The quality of the images needs to be improved. It does not become clear how many tissue specimens were analyzed per sample and how cell counts were performed or how the algorithm was trained to apply Mackay criteria properly. Particularly the whole slides images in Fig. 1, 2 and 3 upper panel are not acceptable. What are the numbers 1-6 in the lower panel?  What were the positive and negative controlls for e-cadherin and Ki-6z staining in Fig. 4 and 5?

I do not understand Fig- 6. How many samples were included considering the numerous locations mentioned in the histopathological procedures? How many images were analyzed per sample? How many cells per image? What was the total number of entosis? Why did the authors chose Mann-Whitney U test?

Minor concerns:

A spell check is reuired, e.g. entoses vs. entosis. Authors should use the same abbraviations/notation throughout the manuscript. Eg. HER2 vs. Her2 vs. HER-2 or e-cadherin vs. E-cadherin vs. e-kadherin or Ki-67 vs Ki67, etc. Structure of sentences should be improved (particularly in the abstract). Since this article is rather a case report, the clinical presentation of the patinets should be improved. The article would be easier to understand, if the clinical characteristics would be mentioned before the tissue investigations.

spell check required, wording needs to be improved

Author Response

Warsaw, 21.07.2023

We thank the reviewers for their work and comments that helped to improve our manuscript.

In the answer to your general comments we reedited the whole manuscript in the structure by putting the main clinical data before results, giving some information as supplementary data instead of in the main text ( Macroscopic descriptions of materials and whole side images). Due to this substantial changes the figures have new numbers (former fig 7 is now fig 1). We deeply revised English spelling and grammar by a native speaker. Because there were so many changes in English editing we are not able to point them all.

And there is our point-by-point answer letter to your comment:

Reviewer 1:

Comment:

The quality of the images needs to be improved. Particularly the whole slides images in Fig. 1, 2 and 3 upper panel are not acceptable.

Asnwer:

The low resolution whole slide images was removed from the main text, and put in supplementary data. The middle and lower panels of Fig 1,2,3 were improved to resolution 300dpi and pasted as tiff bitmaps.

Comment:

What are the numbers 1-6 in the lower panel?

Answer:

They show Mckay’s criteria. Description is given in the figure legenda: numbers 1-6 as follow: 1 and 2) Semi-crescent nucleus of the engulfing cell; 3) cytoplasm of the outer cell; 4) entotic vacuole; 5) cytoplasm of the inner cell; 6) nucleus of the internalized cell.

Comment:

  What were the positive and negative controls for e-cadherin and Ki-6z staining in Fig. 4 and 5?

Answer:

We added: The immunohistochemistry procedures were performed as described in [17]. The controls was done for each slide as follow: The negative control was for E-cadherin - lobular breast cancer for; for Ki67 – skin. The positive control was: for E-cadherin – NOS breast cancer; for Ki76: small cell lung cancer and palatine tonsil. All controls were performed with the same antibodies used within this study. The types and clone numbers of antibodies are given in [17].

Comment:

I do not understand Fig- 6.

Answer:

We changed the figure description (now it is fig 7): The density of entosis (frequency in %) in different localizations of the BC case according to the TNM criteria. T-tumor, N-lymph node, M-metastasis. The calculated density for each TMN localization was: 2.4% (T), 5.8% (N) and 0.75% (M) of entotic figures. Bars represent SEM. The one-way Anova was used in the analysis; * p < 0.001.

Comment:

It does not become clear how many tissue specimens were analyzed per sample and how cell counts were performed or how the algorithm was trained to apply Mackay criteria properly.

Answer:

We added: The histological scans containing > 10,000 cells were assessed by three independent medical doctors, including a certified clinical pathologist. The number of entotic figures was counted using diagnostic criteria established by Mackay [12]. Only figures displaying at least 3 form 5 diagnostic criteria and confirmed by three medical doctors were calculated.

Comments:

 How many samples were included considering the numerous locations mentioned in the histopathological procedures?

 How many images were analyzed per sample?

 How many cells per image?

Answer:

We added: “We analyzed at least 40 randomly captured images per each TMN localizations. Average number of cells per image was 400 (T) and 250 (NM). Thus we analyzed at least 10,000 cells in each TMN localizations.”

Comment:

What was the total number of entosis?

Answer:

Taking into account the number of analyzed cells the total number of entotic figures calculates by us was: 384 (T); 580 (N); 75 (M).

Comment:

Why did the authors chose Mann-Whitney U test?

Answer:

We recalculated statistical significance using proper test: one way ANOVA ( 3 groups) obtaining the similar results.

Comment:

A spell check is required, e.g. entoses vs. entosis.

Answer:

We corrected plural form entoses to either entosis (concerning general process) or entotic figures (concerning many of entoses observed in slides)

Comment:

Authors should use the same abbraviations/notation throughout the manuscript. Eg. HER2 vs. Her2 vs. HER-2 or e-cadherin vs. E-cadherin vs. e-kadherin or Ki-67 vs Ki67, etc. Structure of sentences should be improved (particularly in the abstract).

Answer:

In the revised version we use following abbreviations consequently: HER2. E-cadherin, Ki67

Comment:

 Since this article is rather a case report, the clinical presentation of the patinets should be improved. The article would be easier to understand, if the clinical characteristics would be mentioned before the tissue investigations.

Answer:

Manuscript was submitted as communication. Due to its structure the section materials must be the last one. However, In the revised version we put clinical characteristic at the end on introduction. Thus, figures are renumerated.  

Comments on the Quality of English Language: spell check required, wording needs to be improved

Answer:

We deeply revised English spelling and grammar by the native speaker. Because there were so may changes in English editing we are not able to point them all.

Reviewer 2 Report

The presented paper is of interest since the role of entosis in cancer is still unknown.

The study included only one patient and the difference in frequency of entotic structures was assessed in primary breast  cancer, lymph node metastasis and far metastases.

Since the previous studies reported unfavorable prognosis in entosis-positive patients, the authors expected increasing density of entotic figures in far metastases, but the number of entotic figures was the lowest in far metastases. More studies on larger number of patients are needed to explore the entosis as potential prognostic marker in breast cancer.

1. Methods- the Macroscopic findings section needs editing-only relevant clinical information should be concisely presented

1. Methods-the Hystopathological description/diagnosis need editing end language improvement -only relevant clinical information should be concisely  presented and presented in English (not combining English description with Latin diagnoses)

Author Response

Warsaw, 21.07.2023

We thank the reviewers for their work and comments that helped to improve our manuscript.

In the answer to your general comments we reedited the whole manuscript in the structure by putting the main clinical data before results, giving some information as supplementary data instead of in the main text ( Macroscopic descriptions of materials and whole side images). Due to this substantial changes the figures have new numbers (former fig 7 is now fig 1). We deeply revised English spelling and grammar by a native speaker. Because there were so many changes in English editing we are not able to point them all.

And there is our point-by-point answer letter to your comment:

Reviewer 2

Comment:

Methods- the Macroscopic findings section needs editing-only relevant clinical information should be concisely presented

Answer:

In the revied version macroscopic images are put as supplementary data.

Comment:

Methods-the Histopathological description/diagnosis need editing end language improvement -only relevant clinical information should be concisely  presented and presented in English (not combining English description with Latin diagnoses)

Answer:

The revised Histopathological report is:

Histopathological description of T and N. Invasive cancer NOS, NST NHG 3 (3+3+3)/28 mitoses/10 HPF (field diameter 0.55 cm) and lymphangitic carcinomatosa. Ductal carcinoma in situ foci with high nuclear atypia, constituting about 10% of the cross-sectional area of the tumor, are also present. Extensive foci of necrosis and fibrosis. The tumor infiltrates the skin of the mammary gland, causing ulceration. Pathological classification of the tumor according to pTNM: pT4N3M0.

Histopathological description of the relapse (M): Metastasis of carcinomatous lymph nodes No III/XIX. Cancerous infiltration of the capsule of the lymph nodes and perinodal adipose tissue. In the upper outer quadrant, foci of atypical intraductal hyperplasia were also present. Pathological classification of the tumor according to pTNM: pT4N3M1.
